# Signal Peptide Variants in Inherited Retinal Diseases: A Multi-Institutional Case Series

**DOI:** 10.3390/ijms232113361

**Published:** 2022-11-01

**Authors:** Hiram J. Jimenez, Rebecca A. Procopio, Tobin B. T. Thuma, Molly H. Marra, Natalio Izquierdo, Michael A. Klufas, Aaron Nagiel, Mark E. Pennesi, Jose S. Pulido

**Affiliations:** 1Vickie and Jack Farber Vision Research Center, Wills Eye Hospital, Philadelphia, PA 19107, USA; 2Ocular Genetics, Wills Eye Hospital, Philadelphia, PA 19107, USA; 3Department of Pediatric Ophthalmology and Strabismus, Wills Eye Hospital, Philadelphia, PA 19107, USA; 4Casey Eye Institute, Oregon Health & Science University, Portland, OR 97239, USA; 5Department of Surgery, Medical Sciences Campus, University of Puerto Rico, San Juan, PR 00921, USA; 6Retina Service, Wills Eye Hospital, Philadelphia, PA 19107, USA; 7The Vision Center, Department of Surgery, Children’s Hospital Los Angeles, Los Angeles, CA 90027, USA; 8USC Roski Eye Institute, Keck School of Medicine, University of Southern California, Los Angeles, CA 90033, USA

**Keywords:** signal peptides, inherited retinal dystrophies, in silico prediction

## Abstract

Signal peptide (SP) mutations are an infrequent cause of inherited retinal diseases (IRDs). We report the genes currently associated with an IRD that possess an SP sequence and assess the prevalence of these variants in a multi-institutional retrospective review of clinical genetic testing records. The online databases, RetNet and UniProt, were used to determine which IRD genes possess a SP. A multicenter retrospective review was performed to retrieve cases of patients with a confirmed diagnosis of an IRD and a concurrent SP variant. In silico evaluations were performed with MutPred, MutationTaster, and the signal peptide prediction tool, SignalP 6.0. SignalP 6.0 was further used to determine the locations of the three SP regions in each gene: the N-terminal region, hydrophobic core, and C-terminal region. Fifty-six (56) genes currently associated with an IRD possess a SP sequence. Based on the records review, a total of 505 variants were present in the 56 SP-possessing genes. Six (1.18%) of these variants were within the SP sequence and likely associated with the patients’ disease based on in silico predictions and clinical correlation. These six SP variants were in the *CRB1* (early-onset retinal dystrophy), *NDP* (familial exudative vitreoretinopathy) (FEVR), *FZD4* (FEVR), *EYS* (retinitis pigmentosa), and *RS1* (X-linked juvenile retinoschisis) genes. It is important to be aware of SP mutations as an exceedingly rare cause of IRDs. Future studies will help refine our understanding of their role in each disease process and assess therapeutic approaches.

## 1. Introduction

Signal peptides (SP) are N-terminal extensions of newly synthesized polypeptide chains whose primary function is to target secretory or membrane-bound proteins to and across the endoplasmic reticulum (ER) membrane [1]. Roughly 18% of human proteins in the UniProt (Universal Protein Service) database contain an SP sequence [2]. Characteristically, SP sequences consist of 16 to 30 amino acids (AA) grouped into three chemically well-defined regions: a hydrophilic N-region, a hydrophobic core, and a polar C-region (Figure 1) [1,3]. Each one fulfills an essential role in preprotein processing, including initial interaction with the ER membrane receptors, SP cleavage, and exit from the ER membrane to the cell membrane. SP sequences also contain a cleavage site where the SP is removed from the mature protein once processing is complete. Mutations in the SP sequence can alter the biochemical properties of SPs and result in defects in the co-translational processing of newly synthesized proteins. SP mutations have been found in association with human diseases. In a systematic review, Jarjanzani, et al. identified 26 SP sequence mutations in 21 different genes associated with various human genetic disorders [3].

Inherited retinal dystrophies (IRDs) comprise a heterogenic group of genetically inherited disorders that result in progressive retinal degeneration leading to partial or complete vision loss [4]. The estimated prevalence of IRDs is approximately 1 in 2000 individuals, affecting more than two million individuals worldwide [5]. They commonly display significant variability in genotype, phenotype, and mode of inheritance [4]. The prevalence of SP signal mutations in IRDs is likely very small as there are only a few descriptions of IRD-associated SP mutations in the scientific literature. In one study, Hiroaka, et al. identified a heterozygous 3-bp insertion in the CTG repeat region of exon 1 of the *LRP5* gene in a patient with advanced retinopathy of prematurity (ROP) [6]. Vijayasarathy, et al. analyzed the biochemical consequences of several *RS1* SP variants caused by missense mutations in four different subjects with X-linked juvenile retinoschisis (*XLRS*), and concluded that the mutations affected protein biosynthesis, and resulted in a null *RS1* phenotype [7].

In this study, we herein identify which genes currently implicated in IRDs contain an SP sequence. We then perform a multi-institutional review of genetic testing results to investigate the occurrence of SP variants. We also describe a series of patients with an IRD in whom genetic testing revealed at least one SP sequence variant that was likely contributing to their disease.

## 2. Results

Fifty-six (21%) of the 271 genes currently implicated in IRDs on RetNet possessed an SP sequence (Appendix A). A total of 505 variants were present in the 56 SP-possessing genes, of which six (1.19%) were located within the SP coding sequence and considered disease causing based on our criteria. The SP sequences of the *CRB1* (early-onset retinal dystrophy), *NDP* (familial exudative vitreoretinopathy) (FEVR), *FZD4* (FEVR), *EYS* (RP), and *RS1* (XLRS) genes were affected. Based on SignalP 6.0, one mutation was located within the N-terminal region, three within the hydrophobic core, and two affecting the C-terminal region. Clinical summaries of these six patients are provided in Appendix B. The variants, affected genes, and related IRDs are shown in Table 1. 

The SignalP 6.0 likelihood score, MutationTaster, MutPred, and in silico pathogenicity prediction scores are shown in Table 1. MutationTaster and MutPred classified all six variants as deleterious or pathogenic, respectively. The two exceptions were the *EYS* variant, which could not be evaluated by MutPred because the resulting peptide sequence was shorter than the required 30 residues, and the *RS1* c.52+1 variant, as MutPred cannot evaluate intronic mutations. All six variants had a population frequency of <0.01% in the GnomAD database. 

The three frameshift mutations in *EYS*, *FZD4*, and *RS1* resulted in a SignalP 6.0 likelihood score of 0, reduced from 0.998 in the respective WT sequence. The 21-basepair deletion in the H-region of the *NDP* gene SP sequence also reduced the predicted SP likelihood from 0.991 to zero. The *CRB1* SP variant was not evaluated by SignalP due to the program not being able to readily assess the effects of start codon mutations. The *RS1* c. 52+1 G>C splice site mutation, although classified as deleterious by MutationTaster, did not cause a change in SP likelihood, as it did not result in a peptide sequence change, and the program may not be able to assess the effects of splice site mutations. Cleavage site prediction was lost in all four of the evaluated variants. The *EYS* p.W32L mutation in the non-SP allele obtained a pathogenic score from one predictor, and a likely harmless from the second predictor, while the *CRB1* p.R686C non-SP variant did not achieve disease causing scores from neither predictor.

## 3. Discussion

Signal peptide mutations are an exceedingly uncommon cause of IRDs. There is a paucity of reports of IRDs caused by SP variants. In this study, we identified the 56 genes associated with IRDs that possess an SP sequence and presented six cases of patients with an SP variant implicated in their disease. The unfavorable effects of SP variants depend on the affected region, and the resulting effects on the processing of proteins destined for the secretory pathway.

The N-region is responsible for the initial interaction with the signal recognition particle (SRP) in the ER membrane and plays a role in SP orientation, favoring or preventing translocation across membranes [12]. Positively charged residues, such as lysine and arginine, give this region its characteristically hydrophilic and ionic properties [13]. Missense mutations in these highly conserved positively charged residues can significantly impair the targeting/translocation process to varying degrees due to dysfunctional recognition of the SP sequence by ER membrane receptors [14]. It has also been reported that when the SRP fails to interact with an anomalous SP sequence, the mutated protein is targeted for degradation via activation of the ribosome-associated protein quality control (RAPP) pathway [15,16]. In our study, two compound heterozygous missense mutations in the *CRB1* gene, p.Met1* and p.Arg686Cys, were present in a patient with early-onset retinal dystrophy (EORD). The p.Arg686Cys variant has been previously reported in a compound heterozygous patient with RP [17]. The c.2T>C p.Met1* variant has been previously reported in a Japanese patient with LCA and results in a substitution of the initial methionine residue for threonine [8]. In most cases, mutations affecting the start codon (AUG) are deleterious and result in a null allele [15]. However, translation can rarely initiate in alternate start codons such as those that code for leucine (CUG), although studies show they typically perform at a markedly reduced efficiency compared to AUG codons [16]. In rare cases, downstream AUG codons can also be used for translation initiation. However, this can result in the SP possessing protein accumulating intracellularly, which has been reported as disease causing [18]. A similar *RS1* p.Met1Leu mutation was reported in a patient with XLRS. Cells transfected with mutant cDNA failed to express a mutated RS1 protein due to a blocked translation initiation at the mutant start codon [7]. Although the mutation affects the SP, the role of this variant in our patient’s disease is likely secondary to a near-complete loss of protein production from the affected allele, rather than a loss of SP function alone.

The hydrophobic region is integral to proper SP function as it is involved in conformation and orientation toward the cell membrane, SP cleavage, rate and efficiency of protein translocation, secretion pathway function, and protein processing [19]. We present a 21 base pair deletion in the *NDP* gene, which removed seven amino acids within the hydrophobic core of the SP in a patient with *FEVR*. SignalP 6.0 likelihood and cleavage site prediction scores for this variant decreased to 0. A similar 18-base-pair deletion eliminating six leucine residues within the *LRP5* SP hydrophobic core was reported in a patient with Osteoporosis-Pseudoglioma Syndrome (OPPG) [20]. Functional assays suggested that the mutant *LRP5* polypeptide had impaired ER entry and post-translational processing [20]. Mutations that cause changes in SP hydrophobicity have been shown to reduce protein expression up to 70–90% due to defective targeting to the endoplasmic reticulum, and failed protein translocation [21,22,23]. H-region mutations have also been reported in association with IRDs. Vijasajarthy, et al. evaluated two *RS1* mutations, p.Leu12His and p.Leu13Pro, found in two patients with XLRS *in vitro*. Both histidine and proline have hydrophilic properties, which may disrupt the hydrophobic properties of the SP’s hydrophobic core [24]. In their study, the level of *RS1* protein was nondetectable in the cellular and secreted fractions [7]. However, the study also found that a mutation exchanging one hydrophobic residue for another did not impair SP function. A cell line transfected with an *RS1* p.Leu13Phe plasmid expressed the same *RS1* levels in the culture medium as WT cells [7]. We can postulate that our patient’s disease process involved similar mechanisms of impaired protein ER entry and post-translational processing due to loss of the hydrophobic core region. We also describe two *EYS* p.Met12Aspfs*14 and *FZD4* p.Pro8Argfs*53 frameshift variants found in patients with RP and FEVR, respectively. Although the SP function is likely completely lost, which is supported by the complete loss of SP likelihood prediction in both cases, the entire peptide sequence of the protein is likely affected. Thus, it is more likely that the deleterious effects of the mutation are primarily due to the frameshift mutation. Furthermore, the SP mutations result in truncated EYS and FZD4 proteins, which can also be targeted for rapid degradation by the nonsense-mediated decay (NMD) pathway [25].

Additionally, we found two mutations in the *RS1* gene located within the 5′ donor splice site of intron 1, affecting the SP’s hydrophobic core and C-terminal region. The c.53-859_78+276 variant in case three results in the deletion of exon 2, which contains the AA for the C-terminal region of the SP, leading to a frameshift mutation with a premature stop codon at position 108 (submitted elsewhere as a case report). There was a complete loss of SP likelihood in this variant. As in the *EYS* and *FZD4* mutants, the truncated protein may also undergo degradation via the NMD pathway. *RS1* frameshift mutations that result in truncated proteins have been reported in multiple patients with XLRS [7]. In case five, *RS1* c.52+1 G>C was predicted to be deleterious by one of the in silico predictors. The SignalP 6.0 likelihood score of this variant was unaffected, as the program evaluates the biochemical properties of the entered amino acid sequences, which does not reflect the possible effects of intronic splice site mutations. A similar mutation, c.52+1 G>A, was identified in a Chinese family with XLRS [7]. In vitro analysis of this variant showed an absolute lack of *RS1* protein in transfected cells. This mutation resulted in skipping exon 2, leading to a frameshift at the resulting exon 1 and exon 3 junction and a premature stop codon [7], which may similarly occur in our patient. Deletion of exon 2 removes the C-terminal region, which is responsible for the final cleavage of the SP sequence from the mature protein. Small nonpolar amino acids at positions −1 and −3 from the cleavage site conferred this region an extended beta conformation that provided the peptidase binding site [4,23]. Loss or substitutions of these amino acids are thought to cause failed recognition or cleavage by the signal peptidase. This lead to mutant chains remaining anchored to the microsomal membranes, with eventual removal from the ER by retrograde translocation and degradation, through the endoplasmic-reticulum-associated protein degradation (ERAD) and proteasome pathways [26]. Accumulation of unfolded proteins in the ER can also lead to activation of the unfolded protein response (UPR), which in the presence of continued ER stress can shift from a protective to a proapoptotic pathway, thus causing disease [27,28]. The role of ER stress in IRDs has also been previously established [29,30].

Currently, there are no therapeutic approaches designed specifically for SP sequence mutations. One proposed mechanism is the use of chemical chaperones, which are molecules that bind to the active site of the mutant protein and stabilize or destabilize the folding transition state to compensate for the mutation [31]. In vitro studies have shown that chemical chaperones could potentially correct the abnormal intracellular accumulation of proteins caused by SP mutations, as evidenced by improved clearance of intracellularly trapped hormones, alleviation of ER stress, and reduced cell death [31]. In vitro studies evaluating this approach have shown promising results in diseases such as RP and Fabry disease [26,32,33]. In the absence of a dominant-negative effect caused by an SP variant, where the mutant protein interacts with the WT inside the ER and prevents proper folding and translocation, a gene augmentation approach in which the WT gene is introduced into the cell, or gene editing, could also confer a therapeutic benefit [7].

There are some limitations to our study. Functional in vitro studies were not performed, and the study is retrospective in nature. Therefore, protein processing, secretion, and function could not be assessed. These tests will be helpful in future studies to elucidate the disease mechanisms. Additionally, genetic testing technology has drastically improved over the past decade [34]. Consequently, the development of genomic databases and other resources that are publicly accessible revolutionized our ability to interpret genetic testing data. Therefore, signal peptide mutations may not have been identified in individuals who received testing several years ago due to limitations in technology and reporting. Regarding Signal 6.0, it does not provide a cut-off score for its SP likelihood prediction, which could be helpful in the future in cases where there is a milder loss of likelihood probability. Although SignalP 6.0 can be a useful tool to evaluate SP variants, it may not be able to assess certain types of mutations such as start codon or splice-site mutations, as seen in two of our cases. 

## 4. Materials and Methods

### 4.1. Identification of Signal Peptide Possessing Genes and Cases of Interest

This study was approved by the Wills Eye Hospital (IRB #2021-74) IRB and ethics committee in accordance with the tenants of the Declaration of Helsinki. Genes implicated in IRDs were retrieved from the hereditary retinal disease online database RetNet (https://sph.uth.edu/retnet/ (accessed on 15 April 2021)). The protein sequence database UniProt (https://www.uniprot.org/ (accessed on 16 April 2021)) was then queried to determine which IRD-associated genes contain SP sequences and to identify the positions of the amino acids composing the SP sequences. The most likely locations of the three SP regions (N-terminal, hydrophobic core, and C-terminal) and the cleavage site were then identified using the online signal peptide analysis tool, Signal 6.0 (https://services.healthtech.predcitortu.dk/service.php?SignalP-6.0 (accessed on 20 April 2021)).

Patient genetic results databases from three separate institutions, Wills Eye Hospital, Children’s Hospital of Los Angeles, and Casey Eye Institute, were reviewed to identify cases of patients with both a confirmed IRD diagnosis and a variant located within the SP sequence from June 2011–August 2021. 

### 4.2. In Silico Bioinformatic Evaluation of Signal Peptide Variants

In silico analyses of SP variants were performed using SignalP 6.0, MutationTaster, and MutPred. These freely available online software packages assess the structural and functional biochemical consequences of genetic mutations on proteins. 

The SignalP 6.0 (https://services.healthtech.predcitortu.dk/service.php?SignalP-6.0 (accessed on 20 April 2021)) server uses protein language models containing an extensive list of proteins to predict the presence of SP sequences across all organisms [35]. SignalP 6.0 offers a likelihood score (0–1) of whether an input nucleotide sequence contains an SP based on the biological and structural properties of the amino acid sequence. A 0-likelihood score indicates a 0% chance likelihood that the input sequence contains an SP, whereas a 1-likelihood score indicates a 100% chance of the presence of an SP. Wildtype (WT) sequence retrieved from UniProt and variant amino acid sequences of the identified genes with SP mutations in our patient population were input to SignalP 6.0. Differences in their likelihood scores were expressed as a percentage (%) change. 

MutationTaster (https://www.mutationtaster.org/ (accessed on 1 August 2021)) uses a Random Forest method to predict the disease potential of a mutation and classifies it as either disease-causing or polymorphism. The MutPred suite (http://mutpred.mutdb.org/ (accessed on 1 August 2021)) encompasses a group of web-based tools including MutPred2, MutPred INDEL, and MutPred-Loss of Function. These are sequence-based machine learning models that integrate genetic and molecular information to predict the pathogenicity of amino acid substitutions [36]. All three tools generate a continuous pathogenicity prediction value of 0–1 for a given mutation. Values above 0.70 indicate pathogenicity for missense mutations and values above 0.50 indicate pathogenicity for both in-frame insertion/deletion and frameshift mutations.

The SP variants that were considered as contributing to the disease burden of IRDs in this patient population were those that met all of the following criteria: they were (1) identified in the SP coding sequence, (2) present in genes that have previously been associated with an IRD, (3) present in IRD patients with genetic testing and a clinical phenotype consistent with mutations in the specific gene containing the SP mutation (genotype-phenotype correlation), (4) present in patients without genetic evidence of further mutational burden in the SP-mutated gene in cis outside of the SP region, (5) present in patients without genetic evidence that their IRD may be caused by mutations in other genes, and (6) considered pathogenic or deleterious by at least one of the in silico algorithms.

## 5. Conclusions

In conclusion, we have described several cases of distinct IRDs associated with mutations in the SP of the affected proteins. Previous reports have demonstrated how mutations in the different regions of the SP will impact protein processing and secretion, leading to disease. We hope this information will further expand the awareness of these extremely rare types of genetic mutations and help lay further groundwork for research into the prevalence and role of SP mutations in hereditary ophthalmic diseases and the development of possible therapies for these patients.

## Figures and Tables

**Figure 1 ijms-23-13361-f001:**
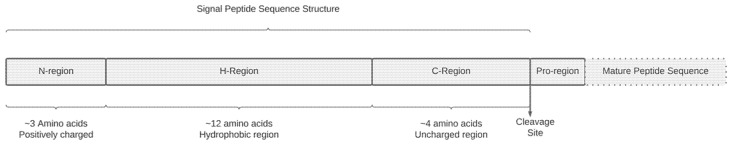
Characteristic SP sequence structure. Classically, the positively charged N-terminal region is composed of 1–5 residues, the hydrophobic core of 7–15 residues, and the uncharged C-region, which contains 3–7 residues. The cleavage site borders the C-region and the pro region.

**Table 1 ijms-23-13361-t001:** Disease-causing SP Variants in Our Patient Population.

Case	Gene	NM	Signal Peptide Mutation	Non-SP Allele Mutation	Diagnosis	Inheritance Pattern	Affected SP Region	Likelihood Change (%) ^a^	Cleavage Site Loss	MutPred ^		MutationTaster		Mutation References
										Sp Allele	Non-SP Allele	SP Allele	Non-SP Allele	
**1**	*CRB1*	201,253.3	c.2T>C (p.Met1*)	c.2056C>T (p.R686C)	retinitis pigmentosa	AR	N-terminal	N/A *	N/A *	0.793	0.465	Disease causing	Polymorphism	Hosono et al. [8]
**2**	*NDP*	_000266	c.37_57del21 (p.L13_M19del)	N/A	familial exudative vitreoretinopathy	XL	H-core	0.991 → 0 (−100%)	Yes	0.619	N/A	Disease causing	N/A	Novel
**3**	*FZD4*	_012193.4	c.23deIC (p.P8Rfs*53)	None	familial exudative vitreoretinopathy	AD	H-core	0.999 → 0 (−100%)	Yes	0.557	N/A	Disease causing	N/A	Novel
**4**	*EYS*	_198283.2	c.32dupT (p. M12Dfs*14)	c.95G>T (p.W32L)	retinitis pigmentosa	AR	H-core	0.999 → 0 (−100%)	Yes	Not performed †	0.872	Disease causing	Polymorphism	Glockle et al. [9] McGuian et al. [10]
**5**	*RS1*	_000330.4	c.52+1 G>C	N/A	X-linked juvenile retinoschisis	XL	C-terminal	0.998 → 0.998 (0%)	No	Not performed ±	N/A	Disease causing	N/A	Vijayasarathy et al. [7]
**6**	*RS1*	_000330.4	c.(52+1_53-1)_(78+1_79-1),del (p.A18Pfs*108)	N/A	X-linked juvenile retinoschisis	XL	C-terminal	0.998 → 0.114 (−88.50%)	Yes	0.529	N/A	Disease causing	N/A	Stone et al. [11]

N/A: Not applicable ^ > 0.70 indicate pathogenicity for missense mutations and >0.50 indicate pathogenicity for both in-frame insertion/deletion and frameshift mutations. * Effects of start codon mutations are not assessed by SignalP 6.0. † MutPred sequence minimum is 30 amino acids. ± MutPred does not assess intronic mutations. ^a^ SignalP 6.0 offers a likelihood score (0–1) of whether an input nucleotide sequence contains an SP based on the biological and structural properties of the amino acid sequence. Differences in likelihood score between the wild-type sequences retrieved from UniProt and variant amino acid sequences of the identified genes with SP mutations in our patient population were expressed as a percentage (%) change.

## Data Availability

Not applicable.

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
