# Peer review of "Signal Peptide Variants in Inherited Retinal Diseases: A Multi-Institutional Case Series"

_ijms, 2022, doi:10.3390/ijms232113361_

Round 1

Reviewer 1 Report

This is a very well-written article reviewing mutations in IRD genes to determine if mutations in signal peptide (SP) sequences are a common cause of disease. This is an interesting hypothesis and one that I don’t think has been addressed elsewhere. Six mutations in SP sequences were identified and analyzed, the conclusion is that SP mutations are infrequent/rare.

The general impression that one gets from the Abstract and Introduction is that the six mutations that were found are likely to cause disease because of their location in the SP sequence. It isn’t until you get to Table 1 that you realize that all but one of the mutations are frameshifts or nonsense mutations and that the most likely mechanism of pathogenicity is simply NMD of the transcripts and loss of function due to no protein being made – the location of the mutation in the SP sequence is likely irrelevant. The one mutation in the NDP gene probably does affect SP function, so the ultimate frequency is 1/505 = 0.2%. The authors do an excellent job of providing alternative explanations for the other five mutations, and always, in the end, admit that the alternatives are not as likely as simple NMD.

Because this was an interesting hypothesis with results that were not under the authors’ control, my only suggestions for improvement involve using stronger adjectives to describe the frequency of SP mutations – “extremely” infrequent, “very” rare, etc.

Additional notes:

Reconsider using examples from the CA gene – it’s unlikely to actually be an IRD gene

In the legend to Table 1, describe where “Likelihood change” comes from

Line 184 – “RS1 frameshift mutations that that result in truncated proteins have been reported…” – insert reference

Reviewer 2 Report

Some errors need to be fixed:

Line 136: The authors report CRB1 c.2T>C (p.Met1*) as a novel variant. Taking into account that the NM used is not provided and assuming they have used the NM of reference (NM_201253.3), the variant CRB1 c.2T>C (p.Met1*) has been previously described in a LCA Japanese patient. (Diagnosis of 34 Japanese Families with Leber Congenital Amaurosis Using Targeted Next Generation Sequencing (Hosono (2018) Sci Rep 8)

In Table 1 should be completed with:

a) Reference NM for each gene

b) A column showing references of previously reported mutations, or indicating if they are novel (as mentioned in the case of the CRB1 mutation)

c) Evaluate the pathogenicity of the Non-SP Allele in CRB1 and EYS. For EYS mutations, it is relevant since the non-SP allele is novel while the EYS c.32dupT (p.M12Dfs*14) is not novel.

d) What is the meaning of N/A, non-available, or non-applicable? Please clarify this point and also write  the  nomenclature meaning in the table footnote.   In case 3, both N/A has the same meaning? For FZD4 gene only dominant inherited pattern has been described.  

Supplementary table1 (Genes associated with iRDs that possess signal peptide sequences and their corresponding chromosomal location, length, and position)

It should be renamed: Genes and mapped loci causing retinal diseases that

possess….

Some names given to genes are wrong, they correspond to loci. i.e: MCDR3 locus.

Genes should be written in italics. Chromosome 5, the correct gene name is ADGRV1 not ADGRV.

Round 2

Reviewer 2 Report

Dear collegues, 

You have incorporated all the suggestions and especially table 1, in my opinion,  is much better now. The phenotype-genotype correlation is the point that I think is missing in this paper. It would be relevant and novel. On the other hand, I understand that it is not easy to obtain the specific clinical data to reach the genotype-phenotype correlation. . Your study is correct and the description of the results has interest for the knowledge of IRDs.